# mRNA Turnover Protein 4 Is Vital for Fungal Pathogenicity and Response to Oxidative Stress in *Sclerotinia sclerotiorum*

**DOI:** 10.3390/pathogens12020281

**Published:** 2023-02-08

**Authors:** Chenghuizi Yang, Lan Tang, Lei Qin, Weiping Zhong, Xianyu Tang, Xin Gong, Wenqi Xie, Yifu Li, Shitou Xia

**Affiliations:** 1Hunan Provincial Key Laboratory of Phytohormones and Growth Development, Hunan Agricultural University, Changsha 410128, China; 2College of Bioscience and Biotechnology, Hunan Agricultural University, Changsha 410128, China

**Keywords:** *Sclerotinia sclerotiorum*, *Ss*MRT4, ribosome assembly, ROS, pathogenicity

## Abstract

Ribosome assembly factors have been extensively studied in yeast, and their abnormalities may affect the assembly process of ribosomes and cause severe damage to cells. However, it is not clear whether mRNA turnover protein 4 (MRT4) functions in the fungal growth and pathogenicity in *Sclerotinia sclerotiorum*. Here, we identified the nucleus-located gene *SsMRT4* using reverse genetics, and found that knockdown of *SsMRT4* resulted in retard mycelia growth and complete loss of pathogenicity. Furthermore, *mrt4* knockdown mutants showed almost no appressorium formation and oxalic acid production comparing to the wild-type and complementary strains. In addition, the abilities to ROS elimination and resistance to oxidative and osmotic stresses were also seriously compromised in *mrt4* mutants. Overall, our study clarified the role of *Ss*MRT4 in *S. sclerotiorum*, providing new insights into ribosome assembly in regulating pathogenicity and resistance to environmental stresses of fungi.

## 1. Introduction

A eukaryotic ribosome, which is considered the “factory” for protein synthesis, is composed of one large (60S) and one small (40S) subunit and a variety of ribosomal proteins (r-proteins). The assembly of ribosomes is a highly precise, strictly regulated process that requires energy consumption and involves a series of factors [1,2,3,4]. There are more than 200 types of eukaryotic ribosome assembly factors, including r-proteins, protein complexes and small nucleolar ribonucleoproteins, which can play a role in different stages such as processing, early assembly, nucleolar to nucleocytoplasmic transport, nuclear remodeling, nucleocytoplasmic transport and ribosome maturation [4]. The abnormal function of ribosome assembly factors may affect the maturation, release, transport and final assembly of ribosome subunits, resulting in serious ribosomopathies and severe damage to cells [5]. Many ribosome assembly factors have been identified in yeast, and their defects will lead to a series of cell metabolism and development abnormalities [6,7,8,9]. Moreover, the deletion of *Bc*Nop53 significantly inhibited its growth and virulence in *Botrytis cinerea* [10]. In *Magnaporthe oryzae*, the ribosome assembly factor *Mo*Fap7 is also involved in mycelial growth and virulence production [11].

mRNA turnover protein 4 (MRT4) is considered to be a transacting factor of ribosome assembly and plays an important role in the maturation of pro-60S subunits of eukaryotic ribosomes. In addition, it also participates in intracellular mRNA turnover [12,13,14,15]. MRT4 was first found during the screening of a yeast mRNA turnover protein, and its mutation causes an mRNA decay defect [15]. Later on, MRT4 was also found in ribosome precursor, which was believed to function in ribosome assembly and maturation [16]. At the initial stage of ribosome assembly, MRT4 and Rpl12 form a complex and anchor together on the ribosome stem ring, affecting the assembly of the ribosome stem [17]. As a collateral homolog of r-protein P0, MRT4 successively interacts with the GAR domain of 25S rRNA, and the replacement process mainly occurs in the cytoplasm [13,18]. Pre-rRNA processing factor Nop53 can target MRT4, participating in the regulation of ribosome assembly [10]. In addition, MRT4 plays an important role in cell tolerance in yeast [19]. The loss of *At*RDP1 (the homolog of MRT4) could decrease the amount of pollen in *Arabidopsis thaliana* and inhibit its development, possibly due to ribosome specialization [20]. It was shown that the subcellular localization of human MRT4 is regulated by the C-terminal region under stress [21]. MRT4 was significantly up-regulated within 15 min when screening the possible genes related to the drug resistance of *Candida albicans* [22], indicating that it may regulate the effectiveness of drugs.

*Sclerotinia sclerotiorum* (Lib.) de Bary, as a pathogenic fungus with a wide host range, mainly colonizes dicotyledons and can seriously interfere with the growth and development of plants [23,24,25,26]. At present, the main method of controlling *S. sclerotiorum* is chemical insecticide control, not only pollutes the environment but also has a poor control effect. With the continuous development of technology, an increasing number of biological control methods have been applied in practice [27,28]. Research on the growth and pathogenesis of *S. sclerotiorum* is conducive to exploring more efficient and environmentally friendly control measures. Most studies of *S. sclerotiorum* focus on its growth and development, oxalic acid synthesis, and secreted proteins [29,30,31]. Recently, research of mycoviruses (or fungal viruses) has gradually increased [32,33,34]; the open reading frame Ι of *Ss*NSRV-1 influences the growth of mycelia and generation of virulence by regulating host protein synthesis pathways [34]. However, it is not clear whether and how ribosome assembly regulates fungal development and infection in hosts. 

In this study, SS1G_11436, which was predicted to be a ribosomal protein, was identified as *Ss*MRT4 in *S. sclerotiorum*. In order to explore the role of SS1G_11436, the main focus of our study is to clarify the biological effects of *SsMRT4* through reverse genetics and provide bases for subsequent research on ribosomal proteins and the comprehensive control of *S. sclerotiorum*. The results show that the mycelia growth of *Ssmrt4* knockdown strains was slow, and more sensitive to stresses. Most importantly, the pathogenicity was completely lost with no appressorium formation and less oxalic acid production in mutant strains, suggesting that *Ss*MRT4 plays a significant role in the formation of appressorium and pathogenicity as well as resistance to oxidative stress in *S. sclerotiorum*.

## 2. Materials and Methods

### 2.1. Fungal Strains and Culture Conditions

A wild-type strain was cultivated on potato dextrose agar (PDA), the knockdown mutants were cultured on PDA with 200 μg/mL hygromycin B (Roche), and the complemented strains were subcultured on PDA with 75 μg/mL G418 Sulfate (Geneticin) (Yeasen). All of them were cultured in an incubator maintained at 20 °C for daily storage.

### 2.2. Plant Materials and Growth Conditions 

Seedlings of *A. thaliana* or *N. benthamiana* used in the experiments were cultured in artificial climate chamber at 22 °C with a treatment of 16 h of light exposure and 8 h of darkness; four-week-old plants were used for further tests.

### 2.3. Identification and Sequence Analysis of SsMRT4

The MRT4 sequences of *S. cerevisiae* (YKL009W), *Fusarium oxysporum Fo47* (EWZ44016.1), *Verticillium dahliae* (XP_009653586.1, RBQ98059.1, PNH42533.1), *B. cinerea B05.10* (BCIN_08g05250), *Pyricularia oryzae 70-15* (XP_003713937.1), *S. sclerotiorum* (SS1G_11436), and *Stagonospora* sp. *SRC1lsM3a* (OAK93896.1) were downloaded from the NCBI database. Multiple sequence alignment of the MRT4 protein sequences was carried out by MEGA7.0 software, and then the neighbor-joining (NJ) method was used to construct a phylogenetic tree. Bootstrap was set to 1000. Protein sequence IDs are shown in the phylogenetic tree. The domains contained in the sequence were analyzed by PredictProtein (https://predictprotein.org/ accessed on 16 February 2022).

### 2.4. Acquisition of Knockdown Mutants and Complementary Strain

The *SsMRT4* gene was knocked down from the genome of *S. sclerotiorum* using the split-marking approach. Primers: ss07gUPF (GACACCTCCCGATTTATTCA)/UR: ss07gUPR (GTGCTCCTTCAATATCATCTTCTCGAGCTTGCGATAGGTAGTG) were designed to amplify nearly 1000 bp of 5′ upstream region of *SsMRT4*. Primers: ss07g DF (CTTGTTTAGAGGTAATCCTTCTTTTTTCCTGAGTGCTATGCC)/DR: ss07g DR (CGGTTACGCATTTGTTGTT) were designed to amplify the 3′ downstream region of *SsMRT4* with nearly 1500 bp. Fragment 1 was composed of the 5′ upstream region of *SsMRT4* and 5′ part of hygromycin phosphotransferase cassette, and fragment 2 was composed of the 3′ downstream region of *SsMRT4* and 3′ part of hygromycin phosphotransferase cassette. These two fragments were then inserted into the T-vector (pEASY^®^-Blunt Cloning Kit, TransGen, Beijing, China). The resulting vector, T-MRT4 was used as a template to amplify two split-marker fragments using primers: *ss07g* UPF(GACACCTCCCGATTTATTCA)/HY-R (AAATTGCCGTCAACCAAGCTC) and YG-F (TTTCAGCTTCGATGTAGGAGG)/*ss07g* DR(CGGTTACGCATTTGTTGTT). These two fragments were co-transformed into wild-type Sclerotinia protoplasts, which could overlap in the hygromycin resistance gene fragment [35]. Primers *ss07g* UPF(GTCTACCTCGTCAAGTCTCCA) and *ss07g* DR (GGACCTATTGAAAGAGTGCG) were used to test whether the *SsMRT4* gene was replaced by the hygromycin-resistant gene. Transformants were purified by hyphal tip transfer at least 3 times. An amplified full-length *SsMRT4* sequence was used to verify mutant strains.

Since we used a split-marker method to generate mutants, in which the target gene was replaced by a hygromycin-resistant gene on site. The mutant is hygromycin-resistant, which may cause difficulties in the subsequent screening of complementary strains if the antibiotic of a complementary vector is also hygromycin. Thus, to reduce the false-positive rate and improve screening efficiency, we changed the hygromycin resistance of the restorer vector pCH-EF-1 (shared by D. Jiang from Huazhong Agricultural University) to G418 Sulfate (Geneticin), named pCH-EF-neo. For *ΔSsMRT4* complementation, the binary vector pCH-EF-neo-MRT4 was constructed using the backbone of pCH-EF-neo. The full-length *SsMRT4* gene, including the promoter and coding sequence (CDS), was amplified from WT genomic DNA. Full-length *SsMRT4* gene fragment and pCH-EF-neo vector were digested using restriction enzyme *Xho*I and *Sac*I, then linked by homologous recombinase (ClonExpress^®^ II One Step Cloning Kit, Vazyme, Nanjing, China) to generate the pCH-EF-neo-MRT4 construct. Then, the plasmid was used for *SsMRT4* transformation via the polyethylene glycol (PEG)-mediated transformation method [35,36].

### 2.5. Analysis of Pathogenecity

To determine pathogenicity, small pieces of mycelia were taken from the edge of PDA medium containing wild-type, *Ssmrt4*-3 and *SsMRT4-C* strains with 2 mm pieces for *A. thaliana* and 5 mm pieces for *N. benthamiana*. Lesion areas were measured 36 h later and counted with Image J. Each experiment was repeated at least three times, and two leaves were used per experiment. The data were analyzed by using SPSS Statistics v.24.0 (IBM, Armonk, NY, USA).

### 2.6. Compound Appressoria Observation and OA Analysis

A 5 mm mycelia plug of *S. sclerotiorum* was placed on a glass slide and cultured for 24 h to observe the formation and number of appressoria. Samples were examined and photographed under stereo microscopes (Stemi508, ZEISS, Oberkochen, Germany) and a light microscope (Axio Imager 2, ZEISS, Oberkochen, Germany). After 16 h of inoculation with *S. sclerotiorum*, onion epidermis was soaked in 0.5% trypan blue solution for 30 min and then decolorized using bleaching solution (ethanol:acetic acid:glycerol = 3:1:1). Samples were examined and photographed under the light microscope (Axio Imager 2, ZEISS).

*S. sclerotiorum* was inoculated on PDA medium containing 100 μg/mL bromophenol blue to detect whether it secreted oxalic acid.

### 2.7. RT-qPCR Analysis

To evaluate the *SsMRT4* expression levels during mycelia development, wild-type strains were cultured on cellophane over PDA, and hyphae were harvested at 1 and 2 days post inoculation (dpi) (hyphae), 3 and 4 dpi (initial sclerotia), 5~7 dpi (developing sclerotia), and 15 dpi (mature sclerotia). Primers of *SsMRT4* and β-tub-ulin used for qPCR were: SsMRT4qF (CCTCCATCATCACCTACTTCC)/SsMRT4 qR: (GGTTCCAAACTAT GTGCCATT) and SsTubqF (ACCTCCATCCAAGAACTC)/SsTubqR (GAACTCCAT CTCGTCCAT). β-tubulin was used as an internal reference. The program setting included holding stage (95 °C, 2 min), cycling stage (95 °C, 20 s; 55 °C, 20 s; 72 °C, 20 s; 40 cycles), and melt curve stage (95 °C, 15 s; 60 °C, 1 min). Quantitative expression assays were performed using SYBR^®^ Green Premix Pro Taq HS qPCR Kit II (Accurate Biology) with StepOneTM Real-time PCR Instrument Thermal Cycling Block. The transcript level of the gene of interest was calculated from the threshold cycle using the 2^-ΔΔ^CT method [37] with three replicates, and data were analyzed using SPSS Statistics v.24.0.

### 2.8. DAB Staining

Using a sterile punch (5 mm), the mycelia-colonized plugs were punched out from the WT, *Ssmrt4-3* and *SsMRT4*-C1 strains of *S. sclerotiorum* (5 pieces each) and placed separately in 5 mL centrifugal tube. Next, 2 mL of 1 mg/mL DAB solution was added into each centrifugal tube, and then the samples were incubated for 30 min at 22 °C in the dark, and immediately photographed (Stemi 508, ZEISS).

### 2.9. Abiotic Stress Response

To test the response of *Ssmrt4*-*3* to cell integrity and different stress, WT, *Ssmrt4*-*3* and *SsMRT4*-*C1* strains were grown on PDA medium with 0.02% SDS, 1 Glucose, 1 M sorbitol, 1 M KCl, 1 M NaCl and H_2_O_2_ (2.5, 5 and 7.5 mM), respectively. After 48 h, the diameter and growth inhibition rate of mycelia were measured: Inhibition rate (%) = 100 × (colony diameter of strain on pure PDA—colony diameter of strain with different stress)/(colony diameter of strain on pure PDA).

### 2.10. Subcellular Localization of SsMRT4

To test the subcellular localization of *Ss*MRT4 in *N. benthamiana*, an *SsMRT4-eGFP* fusion gene driven by a 35S promoter was used for subcellular localization observation. The *SsMRT4-eGFP* constructs were transformed using Agrobacterium GV3101-mediated transformation. The Agrobacterium strains harboring the constructs were used to infiltrate lower epidermal cells of four-week-old *N. benthamiana* leaves [38]. Leaves were examined 48–72 h after infiltration using a Zeiss LSM710 fluorescence microscope. 4′,6-Diamidino-2-phenylindole (DAPI) was used as a nuclear marker. We applied DAPI staining on leaves for 15 min at room temperature. The excitation and emission wavelengths for DAPI were 385 and 420 nm, respectively; and 470–490nm and 500–540 nm for eGFP, respectively.

## 3. Results

### 3.1. Identification of MRT4 in S. sclerotiorum

When *SsMRT4* was used as a query sequence to search for homologs in NCBI database, only one candidate gene (*SS1G_11436*) was identified in *S. sclerotiorum* (Figure 1A). *SS1G_11436* contains a 714-bp ORF with three exons and encodes a protein with a length of 237 amino acids. Phylogenetic tree analysis and sequence alignment showed that SS1G_11436 exhibited a high sequence similarity with *B. cinerea* MRT4 (BCIN_08g05250) (94.51% identity in amino acid sequence) and *S. cerevisiae* MRT4 (YKL009W) (42.26% identity in amino acid sequence) (Figure 1B). Similar to *Bc*MRT4 and *Sc*MRT4, *Ss*MRT4 also contains a Ribosomal_L10 domain as predicted (Figure 1C). To clarify the subcellular localization of *Ss*MRT4, a C-terminal eGFP tag was fused to its coding sequence (*Ss*MRT4-eGFP) and transient transformed into the *N. Benthamian*. As expected, the fused *Ss*MRT4 was located in the nucleus and co-localized with DAPI staining (Figure 1D). 

### 3.2. Knockdown and Complementation of SsMRT4 in S. sclerotiorum

To explore the possible function of *SsMRT4*, we used a split-marker method to generate mutants (Figure 2A), in which target gene was replaced by hygromycin-resistance gene in site. The homozygous knockout mutant, however, stopped growing within 24 h after inoculation, indicating the lethality of *SsMRT4* knockout homozygotes. After continuous purification, we obtained three knockdown strains, *Ssmrt4*-*1*, *2, 3*, and the expression of *SsMRT4* was then detected in these strains. We chose *Ssmrt4-3* with the lowest expression of *SsMRT4* for the subsequent experiments according to qRT-PCR results (Figure 2B). Then, a genetic complementation test was conducted through agrobacterium-mediated transformation. Subsequently, the expression of *SsMRT4* were detected in these *Ssmrt4-3* complementary strains, and *SsMRT4*-*C1* with the highest expression was selected for further experiments (Figure 2B).

When the expression patterns of *SsMRT4* during different developmental stages were determined though qRT-PCR, the results showed that *Ss*MRT4 was highly expressed during the development of sclerotia stage (Figure 2C). However, we found that the *Ssmrt4*-3 phenotype was significantly different from wild-type strain in the process of mycelial culture. While wild-type mycelium grew continuously and normally on the surface of PDA, *Ssmrt4* knockdown mutants cannot form continuous growth of hyphae on the surface, showing a truncated growth state (Figure 2D). At the same time, the growth rate of hyphae in *Ssmrt4* mutants was significantly lower than that of wild-type strains, which were only 0.55 cm/24 hpi (Figure 2E,F). In addition, the number of sclerotia of *Ssmrt4-3* was also significantly smaller than that of wild-type strains (Figure 2G,H). On the other hand, the mycelial phenotype and growth rate of the complementary strains were found to be consistent with those of the wild-type strains, indicating that *SsMRT4* knockdown is responsible for the mutant phenotype. 

### 3.3. SsMRT4 Is Required for Fungal Pathogenicity of S. sclerotiorum

To examine whether *Ss*MRT4 is related to the pathogenicity of *S. sclerotiorum*, we inoculated WT, *Ssmrt4-3* and *SsMRT4-C1* on detached leaves of *A. thaliana* and *N. Benthamian* (Figure 3). Under the same infection conditions, leaves inoculated with WT formed obvious necrotic lesions, whereas leaves infected by *Ssmrt4-3* did not show any necrosis after 36 h. The same results were observed on undetached leaves in *A. thaliana* (Figure 3A,C) and *N. benthamiana* (Figure 3B,D), indicating that *Ss*MRT4 is essential for pathogenicity in *S. sclerotiorum.*

### 3.4. SsMRT4 Contributes to Compound Appressorium Formation and Oxalic Acid Production

The formation of compound appressorium is a key factor for the pathogenicity of *S. sclerotiorum* [39,40]. Compound appressoria formation of WT, *Ssmrt4-3* and *SsMRT4-C1* was determined on the slide surface after incubation for 16 h. Under the stereo microscopes, compound appressorium could not form at all in *Ssmrt4-3* (Figure 4A). After being magnified ten times under the optical microscope, the shape of the compound appressorium can be clearly observed in WT, but no compound appressorium was formed in *Ssmrt4-3* (Figure 4B). When the onion epidermis staining with trypan blue was made 16 h after infection, it was found that *Ssmrt4-3* could grow a small number of mycelia in onion epidermis, but still could not form normal compound appressorium (Figure 4C).

*S. sclerotiorum* can secrete oxalic acid to change the pH value during infection, which is more conducive to the process of infection [41]. We inoculated the mycelial plugs on the PDA medium which contained bromophenol blue to detect the oxalic acid secretion of WT, *Ssmrt4-3* and *SsMRT4-C1*. The results show that the PDA inoculated with WT and *SsMRT4-C1* turned from blue to yellow, indicating that they could produce oxalic acid normally. However, the color of the medium inoculated with the mutant did not change, suggesting that the knockdown of *SsMRT4* means that it is unable to produce acid in *S. sclerotiorum* (Figure 4D). Thus, *Ss*MRT4 plays a very important role in the formation of the appressorium and the production of oxalic acid.

### 3.5. SsMRT4 Is Vital for the Oxidative Stress Response

Ribosome assembly factors may play a role in the fungal response to oxidative stress [11]. In order to test this possibility, we inoculated the fungi plugs on PDA containing hydrogen peroxide at different concentrations. With the increase in hydrogen peroxide concentration, the growth of WT and *SsMRT4-C1* strains slowed down, but they could still grow on PDA medium containing 7.5 mM H_2_O_2_. The *Ssmrt4-3* strain, however, could not grow on the medium under oxidative stress (Figure 5A,C). Then, we used 3,3′-diaminobenzidine (DAB) to stain WT, *Ssmrt4-3,* and *SsMRT4-C1*. After 30 min, we found that only a few wild-type hyphae were stained light brown, while more places on the surface of the *Ssmrt4-3* block were stained dark brown (Figure 5B). It is demonstrated that the H_2_O_2_ content in *Ssmrt4-3* mycelia was higher than in wild-type mycelia, indicative of a vital role of *Ss*MRT4 under oxidative stress in *S. sclerotiorum*.

### 3.6. SsMRT4 Is Essential for the Cellular Integrity of Hyphae under Stresses

To explore the role of *Ss*MRT4 in cell integrity, we inoculated *Ssmrt4-3*, wild-type and complementary strains on a PDA medium that contained 1 M glucose, 1 M sorbitol, 1 M NaCl, 1 M KCl and 0.02% SDS, respectively. It was found that the growth of *S. sclerotiorum* was inhibited under both salt ion stress or high osmotic pressure stress. The inhibition caused by salt ion stress was more severe; *Ssmrt4-3* could not grow at all (Figure 6). Similarly, the growth of *Ssmrt4-3* was completely inhibited on the PDA medium containing 0.02% SDS (Figure 6). Thus, the mutation of *SsMRT4* leads to more sensitivity to hyperosmotic stress and cell integrity perturbation in *S. sclerotiorum*.

## 4. Discussion

In eukaryotes, the large subunit and small subunit are the two parts necessary for the formation of mature ribosomes in all organisms [2,42]. The ribosome stalk, as the key structure of a large subunit, is necessary for recruitment of translation factors and is crucial for ribosome activity [43,44]. RPP0 (or P0) contains the Ribosomal_L10 domain, a direct homologue of L10 protein in prokaryotes and an important component of ribosome stalk in eukaryotes [45]. The nucleolar protein MRT4 is closely correlated with P0 and contains the Ribosomal_L10 domain. Although SS1G_11436 has only 42.26% amino acid sequence identity with *ScMRT4*, as the only *S. sclerotiorum* gene in the phylogenetic tree, it also contains the Ribosomal_L10 domain; therefore, we regard it as *Ss*MRT4. MRT4 is highly conserved in eukaryotes, and exists in the pre-60S ribosome complex rather than the mature 60S subunit. It mainly functions in the nucleolus and nucleoplasm [46]. We transiently expressed *Ss*MRT4 in *N. Benthamian*, and the eGFP signal shows that *Ss*MRT4 was located in the nucleus, which is consistent with previous results. The deletion of Yvh1 leads to a change in the subcellular localization of MRT4 in yeast [17], whether the same result will occur in *S. sclerotiorum* requires more experiments to verify. In order to explore the specific role of *SsMRT4* in *S. sclerotiorum*, we replace the *SsMRT4* with a hygromycin gene in situ to obtain the mutant strains. Despite the fact that the deletion of MRT4 in yeast does not affect the normal growth of yeast [18], our experimental results show that the complete deletion of *SsMRT4* will lead to the premature death of *S. sclerotiorum.* We speculate that this may be due to differences between species, and the complete deletion of *SsMRT4* led to an abnormal ribosome structure that could not normally translate proteins needed for the growth of *S. sclerotiorum*. Fortunately, three knockdown strains were obtained after continuous purification. Thus, we chose the *Ssmrt4*-3, whose expression level of *SsMRT4* was reduced 50 times compared to the WT in the follow-up study. Although the expression of *SsMRT4* has no clear spatiotemporal specificity in the life cycle of *S. sclerotiorum*, except that it is at the stage of sclerotia development on the seventh day after inoculation on PDA, the growth rate of *Ssmrt4-3* is still only a quarter of the wild-type strain. A possible explanation for this might be that *SsMRT4* does not need to maintain high expression but is essential for hyphal growth. In addition, we observed that the mutant could hardly form continuous and normal hyphae on the surface of PDA. Corresponding to the high expression of *SsMRT4* in the 7dpi and final sclerotia, the number of sclerotia in the *Ssmrt4-3* was significantly reduced compared with that of the wild-type strains.

Although the role of MRT4 in fungal infection has not previously been reported, MoFap7, which is also a ribosome assembly factor, has significantly decreased its virulence after being knocked out [11], indicating that ribosome assembly factor may play a role in the formation of fungal virulence. Our results show that after 36 h of infection with *Ssmrt4*-3, no necrotic lesions appeared on plant surfaces. Then, we extended the infection time and found that the pathogenicity of *Ssmrt4*-3 is still completely lost after 4 days of inoculation (Appendix A). These studies suggested that *Ss*MRT4 is essential for the pathogenicity of *S. sclerotiorum*. In order to explore the cause of lost pathogenicity in *Ssmrt4*-3, we examined the formation of its appressorium and the production of oxalic acid. We found that *Ssmrt4*-3 could not form appressorium on either glass or onion surface, and no oxalic acid was produced, indicating that the loss of pathogenicity of the mutant is due to its inability to form appressorium and secrete oxalic acid during infection. In the process of resisting the invasion of *S. sclerotiorum*, host plants will produce a series of immune reactions, including the massive production of ROS [47,48]. As an important part of ROS, H_2_O_2_ was added to PDA to simulate this process. It was found that even the lowest concentration of H_2_O_2_ made *Ssmrt4-3* completely unable to grow. This may be caused by the loss of hydrogen peroxide scavenging ability in the mutant, and led to a high ROS accumulation that inhibited the growth of mycelia. At the same time, the H_2_O_2_ content in the *Ssmrt4-3* under normal culture conditions was also significantly higher than that in the wild-type strain, which further verified our previous hypothesis.

In conclusion, we identified *SsMRT4* and created knockdown mutants of *SsMRT4* in *S. sclerotiorum*. Based on the above results, we established a model (Appendix A) to reveal the function of *SsMRT4* in *S. sclerotiorum*. Firstly, *Ss*MRT4 influences the accumulation of pre-60S subunits as a ribosomal transacting factor, thereby affecting the assembly of immature ribosomes in the nucleus. The damage of ribosome assembly further affects the synthesis of proteins in *S. sclerotiorum*. The disruption of protein synthesis directly affects the hyphal growth and integrity of ROS clearance pathway in *S. sclerotiorum*. Secondly, both endogenously produced and exogenously accumulated ROS cannot be eliminated in time, which further affects the growth and development of *S. sclerotiorum*. Finally, the failure to form mature proteins seriously interferes with the formation of appressorium and secretion of oxalic acid, resulting in the complete loss of pathogenicity of the mutant.

## Figures and Tables

**Figure 1 pathogens-12-00281-f001:**
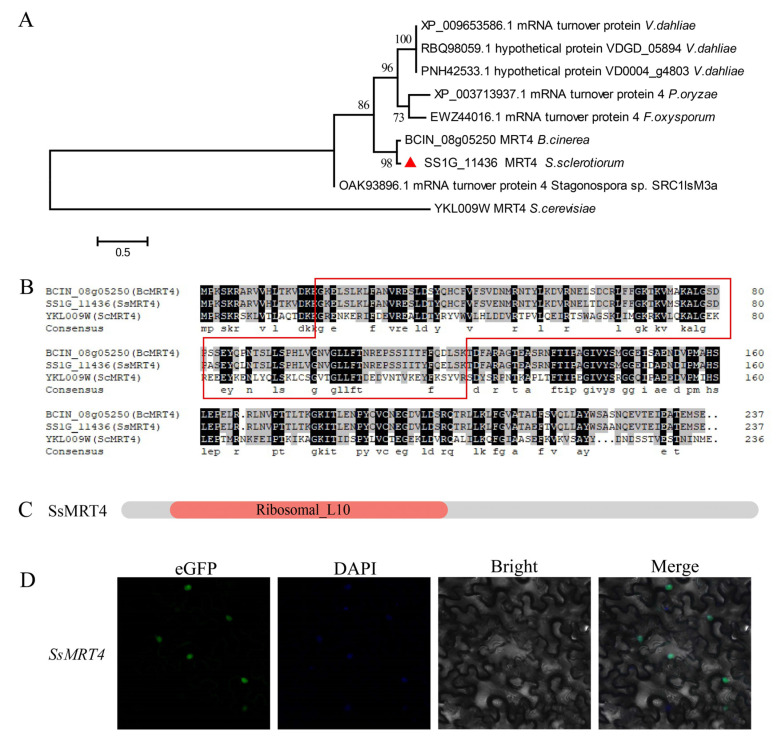
Identification and subcellular localization of *Ss*MRT4. (**A**) Phylogenetic tree of *Ss*MRT4 was constructed based on amino acid sequences from different fungi. Data were downloaded from EnsemblFungi (http://fungi.ensembl.org/index.html, accessed on 28 December 2022). (**B**) Alignment of amino acid sequences of *B. cinerea* MRT4 (BCIN_08g05250), *S. cerevisiae* MRT4 (YKL009W) and *Ss*MRT4 (SS1G_11436). The predicted domain sequence is highlighted in red box. (**C**) Protein domain structure analysis. Domain architectures were identified using Pfam database (https://pfam.xfam.org/, accessed on 28 December 2022) (**D**) *Ss*MRT4 protein subcellular localization in *N. Benthamian.* Merge: merged by eGFP and DAPI.

**Figure 2 pathogens-12-00281-f002:**
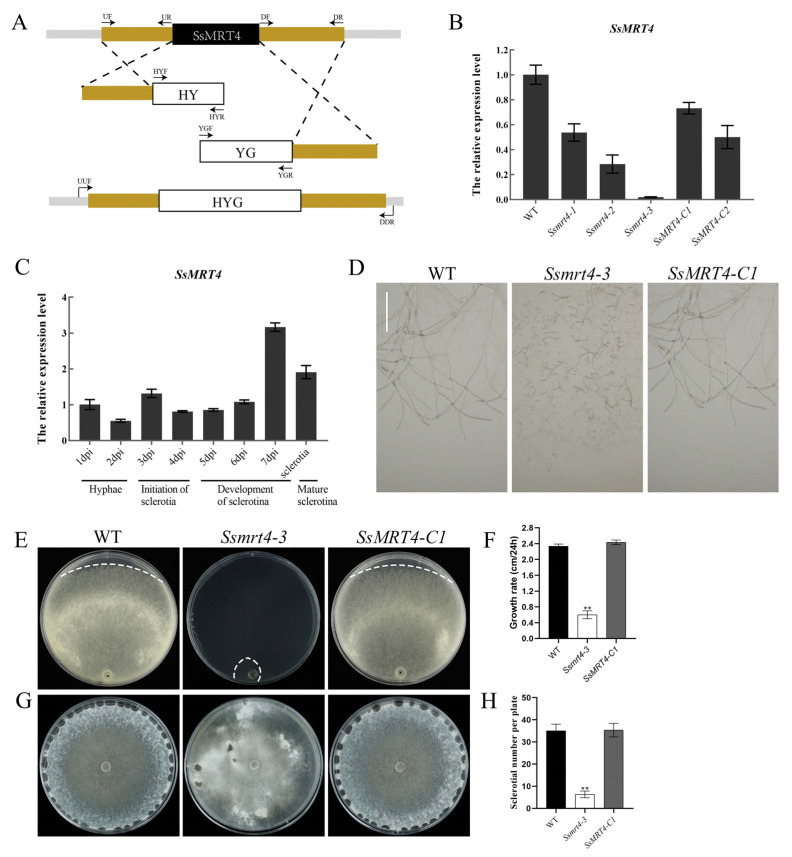
Phenotype of *SsMRT4* knockdown and complementary strains. (**A**) Technical process of split marker method. Fragment 1 consists of the upstream part of the gene of interest and the first half of the hygromycin resistance (HY) gene, while fragment 2 consists of the latter half of hygromycin and the downstream part of the gene of interest. These two fragments replaced the gene of interest by homologous recombination with the hygromycin resistance gene. (**B**) Relative expression level of *SsMRT4*. β-tubulin was used as an internal reference. Relative expression of *SsMRT4* at 1 dpi was set as control. WT, wild-type strain; *Ssmrt4-1/2/3*, knockdown strains; *SsMRT4-C1/C2*, complementation strains. (**C**) Relative expression levels of *SsMRT4* during different developmental stages of sclerotia. Error bars represent SD. (**D**) Hyphae morphology of WT, *Ssmrt4*-*3* and *SsMRT4*-*C1* on PDA. Bar = 1 mm. (**E**) Semidiameter of mycelial growth at 40 h. The white dotted line is the edge of mycelial growth. (**F**) Growth rate of WT, *Ssmrt4-3* and *SsMRT4-C1*. (**G**,**H**) Sclerotial number per plate of WT, *Ssmrt4-3* and *SsMRT4-C1*. Experiments were conducted three times with similar results. Error bars represent SD. Statistical significance was analyzed using Student’s *t*-test between wild-type and knockdown mutants or complementation strains (** *p* < 0.01). WT, wild-type; *Ssmrt4-3*, knockdown strain; *SsMRT4-C1*, complementation strain.

**Figure 3 pathogens-12-00281-f003:**
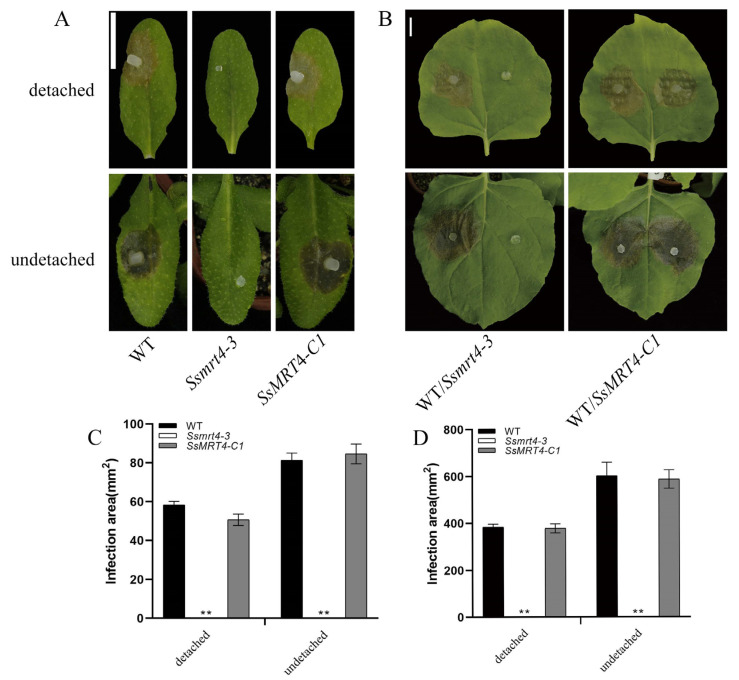
*Ss*MRT4 is required for *S. sclerotiorum* pathogenicity. (**A**,**C**) Inoculated lesions of WT, *Ssmrt4-3* and *SsMRT4-C1* of detached and undetached leaves of *A. thaliana* and *N. benthamiana*. Data were recorded at 36 hpi. Bar = 10 mm. (**B**,**D**) Lesion areas of WT, *Ssmrt4-3* and *SsMRT4-C1* on leaves of *A. thaliana* and *N. benthamiana*. Image J was used to analyze lesion areas. Experiments were conducted three times with similar results. Error bars represent SD. Statistical significance was analyzed using Student’s *t*-test between wild-type and knockdown mutants or complementation strains (** *p* < 0.01). WT, wild-type; *Ssmrt4-3*, knockdown strain; *SsMRT4-C1*, complementation strain.

**Figure 4 pathogens-12-00281-f004:**
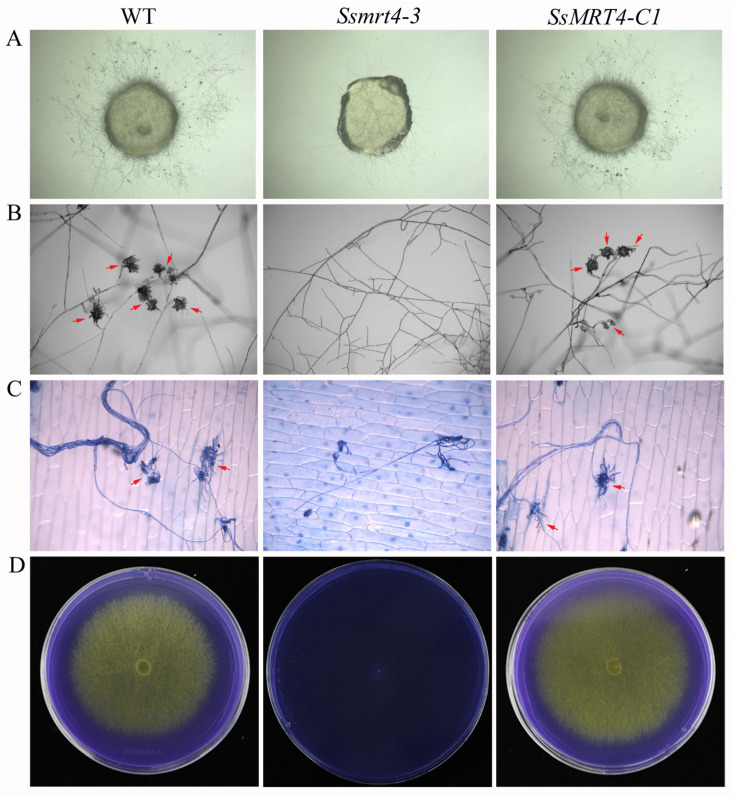
*Ss*MRT4 affects appressorium formation and oxalate secretion. (**A**) WT, *Ssmrt4-3,* and *SsMRT4-C1* were placed on glass slides and cultured for 24 h to observe formation and number of appressoria under the stereo microscopes. (**B**) Compound appressorium observation of WT, *Ssmrt4-3* and *SsMRT4-C1* under the optical microscope. Red arrows: compound appressorium. (**C**) Penetration assay of WT, *Ssmrt4-3,* and *SsMRT4-C1* on onion epidermis cells. Invasion mycelial were stained by trypan blue. Red arrows: compound appressorium. (**D**) Mycelium of WT, *Ssmrt4-3* and *SsMRT4-C1* grown on PDA medium containing bromophenol blue. Experiments were conducted three times with similar results. WT, wild-type; *Ssmrt4-3*, knockdown strain; *SsMRT4-C1*, complementation strain.

**Figure 5 pathogens-12-00281-f005:**
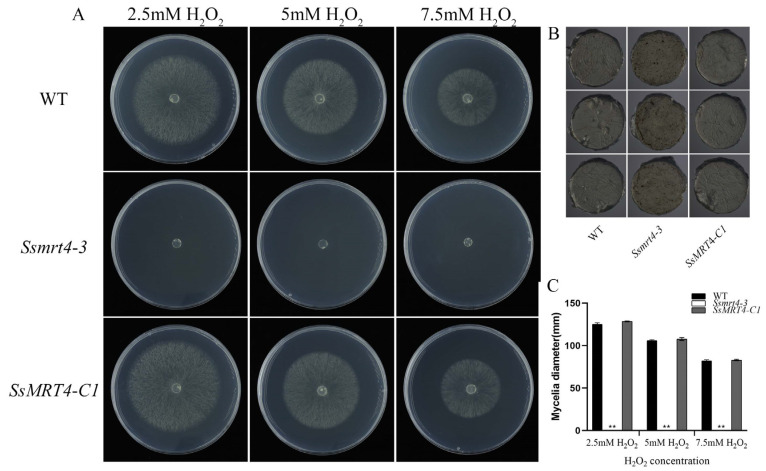
*Ss*MRT4 affects ROS elimination. (**A**) Colonial morphology and mycelial growth of WT, *Ssmrt4-3* and *SsMRT4-C1* under different concentrations of H_2_O_2_. (**B**) DAB staining of WT, *Ssmrt4-3* and *SsMRT4-C1*. (**C**) Hyphae growth of WT, *Ssmrt4-3* and *SsMRT4-C1*. Experiments were conducted three times with similar results. WT, wild-type; *Ssmrt4-3*, knockdown strain; *SsMRT4-C1*, complementation strain. Error bars represent SD. Statistical significance was analyzed using Student’s *t*-test between wild-type and knockdown mutants or complementation strains (** *p* < 0.01).

**Figure 6 pathogens-12-00281-f006:**
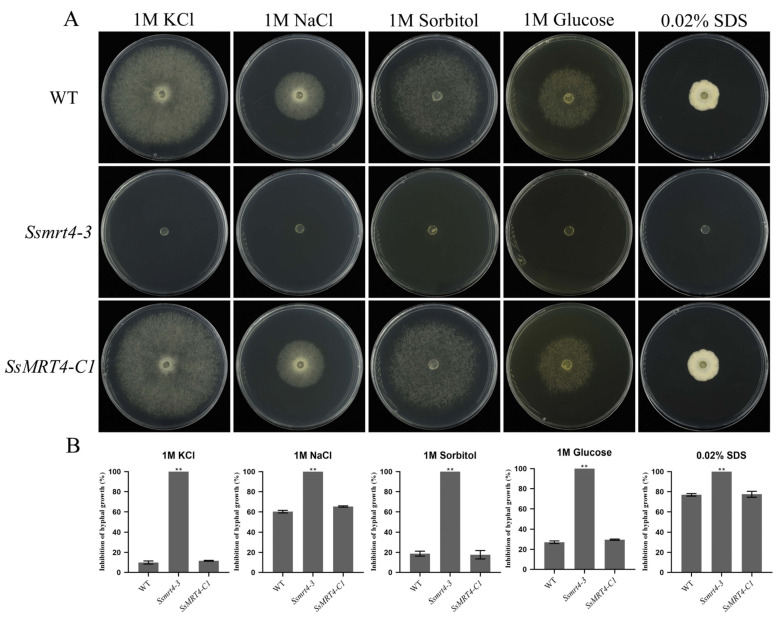
*Ss*MRT4 is essential for cellular integrity of hyphae. (**A**) Sensitivity of *Ssmrt4-3* to hyperosmotic stress and cell integrity perturbation agents. *Ssmrt4-3* mutant was inoculated on PDA plates amended with 1 M sorbitol, 1 M glucose, 0.02% sodium dodecyl sulphate (SDS), 1 M NaCl and 1 M KCl, respectively. The inhibition of hyphal growth was then calculated at 48 hpi. (**B**) Growth inhibition rate of *Ssmrt4*-3. Error bars represent SD. Statistical significance was analyzed using a Student’s *t* test between wild-type strains and each mutant (** *p* < 0.01). WT, wild-type strain; *Ssmrt4-3*, knockdown strain; *SsMRT4-C1*, complementation strain.

## Data Availability

Not applicable.

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
