# Peer review of "mRNA Turnover Protein 4 Is Vital for Fungal Pathogenicity and Response to Oxidative Stress in Sclerotinia sclerotiorum"

_pathogens, 2023, doi:10.3390/pathogens12020281_

Round 1

Reviewer 1 Report

Dear Authors

Overall, the manuscript provides a nice study in mRNA turnover protein 4 is vital for fungal pathogenecity and response to oxidative stress in Sclerotinia sclerotiorum. However, the reviewer thinks there are some aspects that can be improved answered as listed below:

1.     The introduction could be improved and more focused on new researches in this field.

2.     This reviewer thinks that an additional section in relation to novel aspects of this work.

3.     Please write the aims of this research, clearly.

4.     Add these new papers in introduction or discussion sections:

Encapsulation of Plant Biocontrol Bacteria with Alginate as a Main Polymer Material

A novel encapsulation of Streptomyces fulvissimus Uts22 by spray drying and its biocontrol efficiency against Gaeumannomyces graminis, the causal agent of take-all disease in wheat

Reducing Drought Stress in Plants by Encapsulating Plant Growth-Promoting Bacteria with Polysaccharides

Finally, after these minor corrections, this manuscript can be publish in Pathogens journal. Please send me the revised version before publishing.

Author Response

I am very grateful to your constructive comments for the manuscript. We have  made modification on the original manuscript, according to your comments and advices. Pls see the details of our responses to these comments listed below.

Author Response

(The authors gave the same response as above.)

Reviewer 3 Report

 The authors describe the role of SsMRT4 in appressoria formation, formation of oxalic acid and the response to different stresses. The study is well designed, the results are clearly presented and the discussion is clear and comprehensible. I have only a few minor remarks. I recommend to accept the manuscript after minor revision. Congratulation to the authors!

Notes:

please check the word separations in the whole manuscript! e.g. 100-101: fragment

79: were cultured

85-87: please indicate the accession of the respective protein sequence

Figure 2C: please use a larger font for the captions below the graph

246: N. benthamiana

Figure 5C + 6B: Please increase the font size of the captions

305: which contained

338: complete deletion

344: compared to the WT

363: pathogenicity

374: please show the model you developed in a figure. This way it will be easier for the reader to retrace your results

Author Response

(The authors gave the same response as above.)

Round 2

Reviewer 1 Report

Accept